# C-mannosylation supports folding and enhances stability of thrombospondin repeats

**Aleksandra Shcherbakova[1], Matthias Preller[2], Manuel H Taft[2], Jordi Pujols[3], Salvador Ventura[3], Birgit Tiemann[1], Falk FR Buettner[1], Hans Bakker[1]\***

[1]Institute of Clinical Biochemistry, Hannover Medical School, Hannover, Germany; [2]Institute for Biophysical Chemistry, Hannover Medical School, Hannover, Germany; [3]Institut de Biotecnologia i Biomedicina, Universitat Autònoma de Barcelona, Bellaterra, Spain

**Abstract** Previous studies demonstrated importance of C-mannosylation for efficient protein secretion. To study its impact on protein folding and stability, we analyzed both C-mannosylated and non-C-mannosylated thrombospondin type 1 repeats (TSRs) of netrin receptor UNC-5. In absence of C-mannosylation, UNC-5 TSRs could only be obtained at low temperature and a significant proportion displayed incorrect intermolecular disulfide bridging, which was hardly observed when C-mannosylated. Glycosylated TSRs exhibited higher resistance to thermal and reductive denaturation processes, and the presence of C-mannoses promoted the oxidative folding of a reduced and denatured TSR in vitro. Molecular dynamics simulations supported the experimental studies and showed that C-mannoses can be involved in intramolecular hydrogen bonding and limit the flexibility of the TSR tryptophan-arginine ladder. We propose that in the endoplasmic reticulum folding process, C-mannoses orient the underlying tryptophan residues and facilitate the formation of the tryptophan-arginine ladder, thereby influencing the positioning of cysteines and disulfide bridging.

**\*For correspondence:**
bakker.hans@mh-hannover.de

**Competing interests:** The authors declare that no competing interests exist.

## Introduction

Protein glycosylation is a major form of co- and post-translational modification, affecting the majority of secreted and cell-surface proteins. It can influence the folding of a protein as well as its physical properties, activity and ability to interact with other macromolecules, thus playing an important role in a great variety of cellular processes (*Varki, 2017*). C-mannosylation is presently one of the less well characterized glycosylation types. It takes place in the endoplasmic reticulum (ER), presumably co-translationally (*Doucey et al., 1998*; *Krieg et al., 1998*), and is performed by C-mannosyltransferase enzymes of the DPY(dumpy)-19 family (*Buettner et al., 2013*; *Niwa et al., 2016*; *Shcherbakova et al., 2017*). C-mannosylation has been found on tryptophans of WXXW/C motifs and involves a carbon-carbon linkage between the sugar and the protein: the C1 atom of the α-mannose is thereby attached to the indole C2 atom of the tryptophan (*Figure 1A*) (*de Beer et al., 1995*; *Hofsteenge et al., 1994*). However, the function of C-mannosylation remains elusive.

C-mannosylation has been predicted on 18% of human secreted and transmembrane proteins (*Julenius, 2007*). There are two main protein groups bearing conserved C-mannosylation sites – proteins with thrombospondin type 1 repeats (TSRs) and type I cytokine receptors. TSRs are small protein domains consisting of around 60 amino acids. They possess a conserved structure of three antiparallel strands typically linked by three disulfide bridges (*Figure 1B*) (*Tan et al., 2002*). The first strand contains the C-mannosylation motif with up to three tryptophan residues (WXXWXXWXXC), which all can be C-mannosylated (*Hofsteenge et al., 1999*). The tryptophans intercalate with

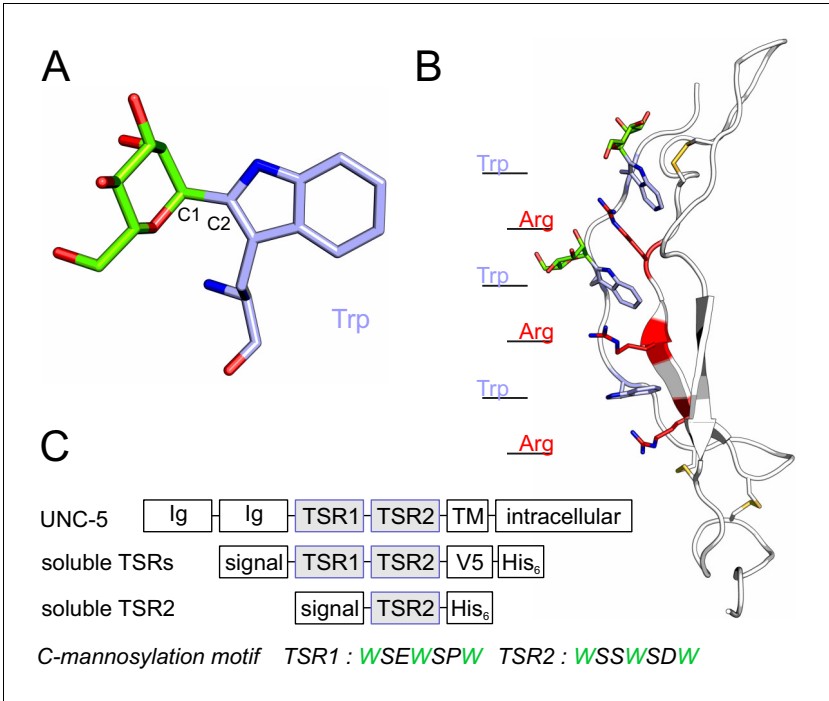

**Figure 1.** C-mannosylation of thrombospondin type 1 repeats. (**A**) C-mannose (green) linked to the indole C2 atom of a tryptophan (light-blue) by a C-C bond. (**B**) Modeled structure of *C. elegans* UNC-5 TSR2 with tryptophans (blue) and arginines (red) arranged in a Trp-Arg ladder. C-mannoses (green) are displayed on the first two tryptophans of the WXXWXXW motif, according to previous findings (***Buettner et al., 2013***), (***Figure 3—figure supplement 1***). Oxygen atoms are indicated in red, nitrogen atoms in dark-blue and disulfide bonds in yellow. (**C**) Natural *C. elegans* UNC-5 and soluble TSR constructs applied in this study. Ig: immunoglobulin-like domain; TSR: thrombospondin type 1 repeat; TM: transmembrane domain; signal: cleavable secretion signal; V5 and His$_6$: tags for detection and purification.

conserved arginine residues from the second strand building a so-called tryptophan-arginine (Trp-Arg) ladder that forms the core of the TSR fold (***Figure 1B***) (***Tan et al., 2002***; ***Tossavainen et al., 2006***). Due to stabilizing cation-π interactions between the arginine and tryptophan side chains (***Gallivan and Dougherty, 1999***), the Trp-Arg ladder is deduced to have an important structural function in the TSRs.

Several studies have shown that C-mannosylation is important for proper secretion of TSR-containing and other proteins. Mutagenesis of the C-mannosylation sites usually resulted in reduced secretion or cell-surface expression of the proteins and their retention in the ER (***Fujiwara et al., 2016***; ***Gouyer et al., 2018***; ***Hilton et al., 1996***; ***Niwa et al., 2016***; ***Okamoto et al., 2017***; ***Sasazawa et al., 2015***; ***Taylor et al., 1997***). The same effects were observed in cells with reduced availability of the donor substrate dolichol-P-mannose (***Perez-Vilar et al., 2004***; ***Wang et al., 2009***), suggesting a direct influence of C-mannose on secretion efficiency; however, other glycosylation processes were affected in these cells as well. With the discovery of the enzyme catalyzing C-mannosylation – the C-mannosyltransferase (***Buettner et al., 2013***) – specific genetic intervention affecting C-mannosylation became possible, enabling to study the effects of C-mannosylation without affecting the target protein sequence or other cellular pathways. This indeed allowed to demonstrate that lack of C-mannosylation alone was responsible for reduced secretion of TSR-containing proteins (***Buettner et al., 2013***; ***Niwa et al., 2016***; ***Shcherbakova et al., 2017***). In this study, we utilized the possibility to produce a single TSR with and without C-mannoses in the same expression system to directly evaluate the effects of C-mannosylation beyond secretion.

## Results

### C-mannosylation becomes critical for secretion of UNC-5 TSRs with increasing temperature

Secretion of TSRs from *C. elegans* netrin receptor UNC-5 (*Figure 1C*) was analyzed at 20, 24°C and 28°C in naturally C-mannose-negative *Drosophila* S2 cells (*Hofsteenge et al., 2001*; *Krieg et al., 1997*). C-mannosylation of the TSRs was obtained by co-expression of the *C. elegans* DPY-19 C-mannosyltransferase. A C-mannosylation-independent Notch EGF16–20 fragment was used as transfection and secretion control.

At 20°C, secretion of UNC-5 TSRs was higher from cells co-expressing DPY-19 than from C-mannosylation-negative cells (*Figure 2*). At increasing temperatures, secretion of C-mannosylated TSRs was not affected, whereas secretion of non-mannosylated TSRs further declined gradually. Intracellular UNC-5 TSR levels were comparable at all conditions confirming that the cells were able to produce C- and non-mannosylated TSRs per se (*Figure 2*). Thus, impaired temperature-dependent secretion of non-mannosylated TSRs implied a role of C-mannoses in protein stability and/or protein folding.

### C-mannosylation increases resistance of UNC-5 TSR2 to thermal denaturation

To explore the impact of C-mannosylation on TSR stability, the second TSR (TSR2) of UNC-5 (*Figure 1B,C*) was produced with and without C-mannoses in *Drosophila* S2 cells. Presence of C-mannoses was verified by mass spectrometry, showing that the major fraction of the protein was modified on the first two tryptophans of the WXXWXXW motif (*Figure 3—figure supplement 1*). The correct folding of both TSR forms was confirmed by circular dichroism (CD) spectroscopy

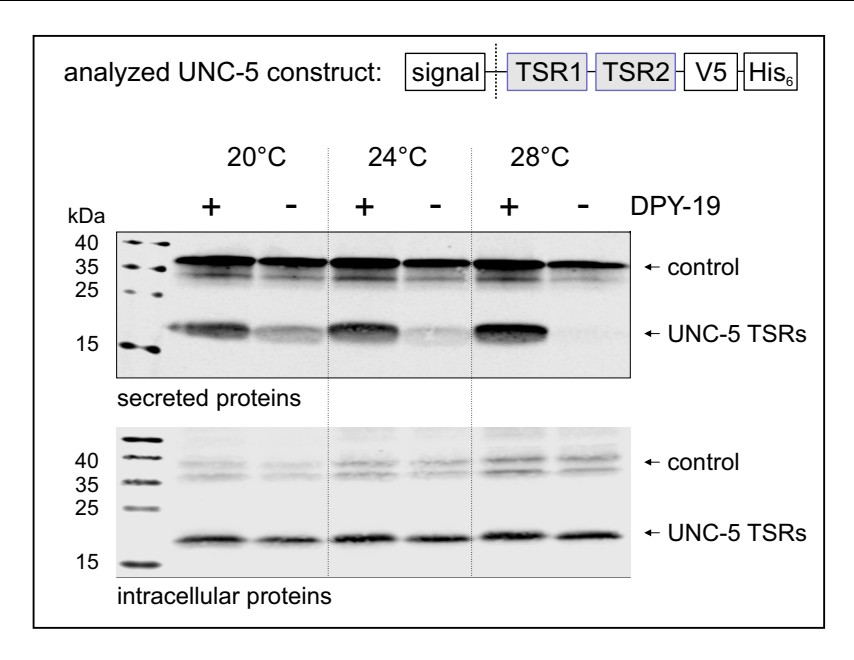

**Figure 2.** Effects of C-mannosylation on TSR secretion. Western blot analysis of secreted (upper panel) and intracellular (lower panel) UNC-5 TSRs 1+2 expressed in *Drosophila* S2 cells, co-transfected with *C. elegans* DPY-19 (+) or an empty vector (-) and incubated at 20, 24°C and 28°C as indicated. V5-tagged EGF repeats 16–20 from *Drosophila* Notch were used as transfection and secretion control. Both proteins were detected by anti-V5 antibody. An analog temperature-sensitivity can be observed in *C. elegans dpy-19* mutants (*Figure 2—figure supplement 1*).

The online version of this article includes the following figure supplement(s) for figure 2:

**Figure supplement 1.** The temperature-sensitive dumpy phenotype of *C. elegans dpy-19* mutants.

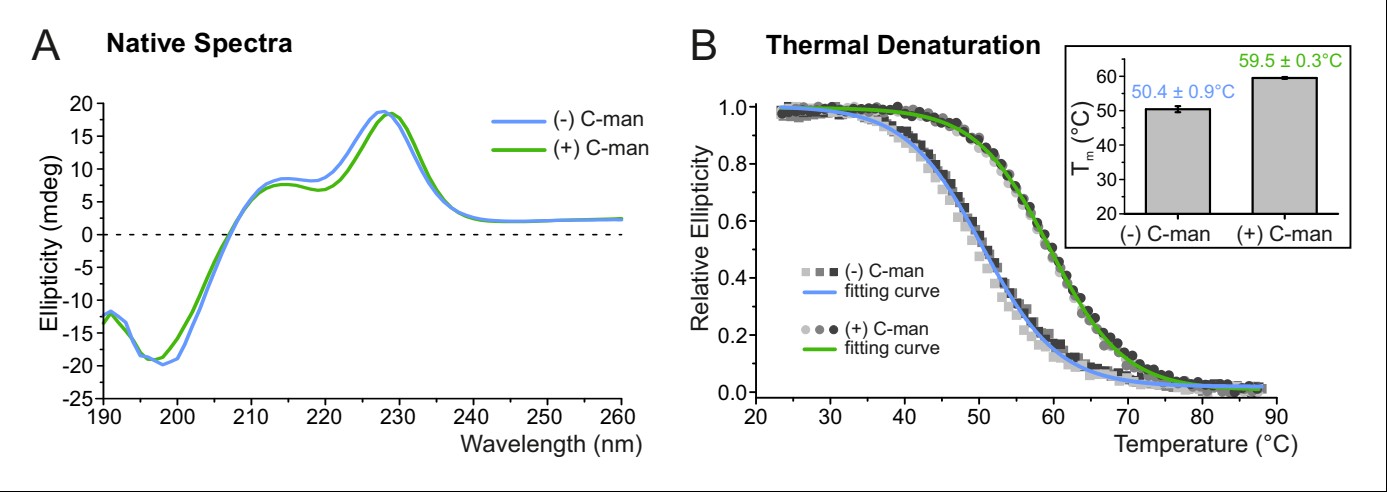

**Figure 3.** C-mannosylation increases resistance of UNC-5 TSR2 to thermal denaturation. (**A**) Native CD spectra of non-mannosylated (blue) and C-mannosylated (green) UNC-5 TSR2 at 24°C. Average spectra of three measurements are displayed for each TSR form (***Figure 3—source data 1***). (**B**) Thermal denaturation of non- and C-mannosylated UNC-5 TSR2 monitored by CD spectroscopy at a wavelength of 229 and 228 nm, respectively. Measurements of three non-mannosylated (gray-shaded squares) and C-mannosylated (gray-shaded circles) UNC-5 TSR2 samples were averaged and fitted (blue and green, respectively) (***Figure 3—source data 2***). Average $T_m$ values are depicted in the right plot (error bars show standard deviation). Thermal denaturation of both, non- and C-mannosylated TSRs, was reversible to a high extent (***Figure 3—figure supplement 2***).

The online version of this article includes the following source data and figure supplement(s) for figure 3:

**Source data 1.** Native CD spectra of C- and non-mannosylated UNC-5 TSR2.
**Source data 2.** Thermal denaturation of C- and non-mannosylated UNC-5 TSR2.
**Figure supplement 1.** Mass spectrometry analysis of UNC-5 TSR2 co-expressed with DPY-19.
**Figure supplement 2.** Recovery of thermally denatured UNC-5 TSR2 with and without C-mannosylation.

showing TSR-typical spectra with characteristic maxima near 230 and 212 nm and a minimum in the 195–200 nm region (***Figure 3A***) (***Huwiler et al., 2002***; ***Roszmusz et al., 2002***; ***Smith et al., 1984***). Disappearance of the 230 nm maximum was used to monitor thermal unfolding of mannosylated and non-mannosylated TSRs (***Figure 3B***). The C-mannosylated TSR revealed a significantly higher melting temperature ($T_m$ = 59.5°C) compared to the non-mannosylated form ($T_m$ = 50.4°C) showing that C-mannosylation on W1 and W2 of the WXXWXXW motif strongly increases the resistance of the TSR to thermal denaturation.

## Molecular dynamics simulation of the thermal denaturation

To understand the structural dynamics behind the increased resistance of the C-mannosylated TSR during thermal denaturation, we performed molecular dynamics simulations of a modeled UNC-5 TSR2 structure with and without C-mannosylation at elevated temperature of 75°C for 200 ns. Each simulation was repeated three times.

In both TSR forms, the Trp-Arg ladder scaffold was mostly maintained during the simulation, presumably because of restricted dynamics due to the three disulfide bridges. However, positions of arginines and tryptophans in the C-mannosylated TSR remained more constrained in all three simulation replicates (***Figure 4A***) indicating that the Trp-Arg ladder structure in the C-mannosylated TSR is more rigid compared to the non-modified TSR. Particularly Trp5 – the first tryptophan of the WXXWXXW motif – appeared highly flexible and diverged from the residual Trp-Arg ladder organization in the non-mannosylated form (***Figure 4—videos 1*** and ***2***). This was confirmed by calculated root-mean-square fluctuations (RMSF) of single residues as depicted by the blue-white-red color gradient of the representative TSR structures from each simulation replicate (***Figure 4B***, ***Figure 4—figure supplement 1***). Correspondingly, calculated cation-π interactions between Trp5 and Arg24 were found to be maintained in the C-mannosylated TSR but not in the non-modified TSR (***Figure 4—figure supplement 2***).

Furthermore, we found hydrogen bond interactions between C-mannoses and the neighboring side chains as well as the backbone of the corresponding tryptophan residues throughout the

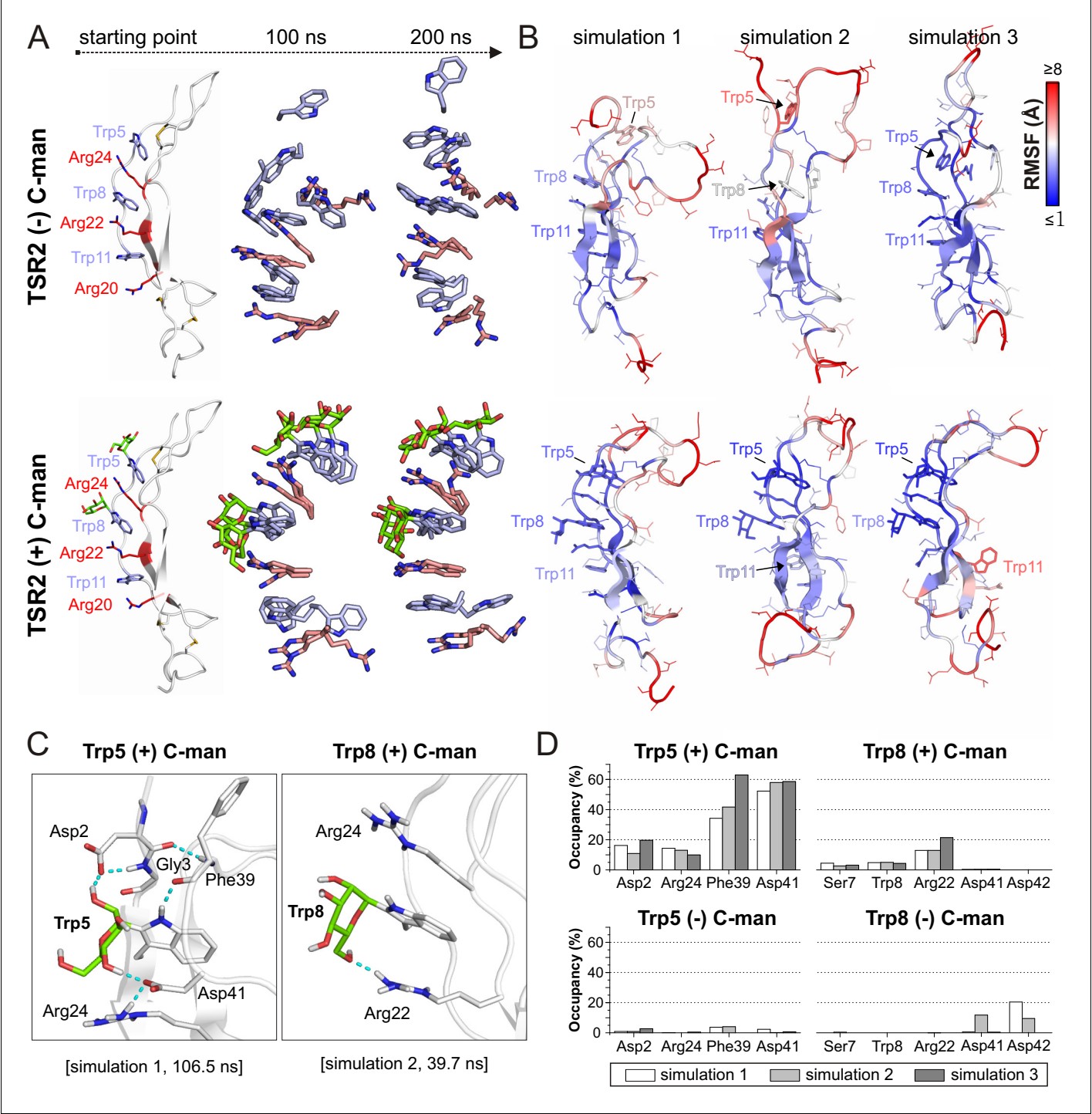

**Figure 4.** Molecular dynamics simulations at elevated temperature. (A) Modeled structures of UNC-5 TSR2 without C-mannosylation (top) and with two C-mannoses on Trp5 and Trp8 (bottom) at the starting point of each simulation followed by aligned Trp-Arg ladder structures from three simulation replicates after 100 and 200 ns simulation time. Tryptophans are depicted in light-blue, mannoses in green and arginines in salmon. (B) Representative most-abundant structures of non- and C-mannosylated UNC-5 TSR2 from each simulation, colored using a blue-white-red gradient according to the calculated root-mean-square fluctuation (RMSF) values (blue ≤1 Å; red ≥8 Å) (*Figure 4—figure supplement 1*, *Figure 4—source data 1*). (C) Hydrogen bonds (cyan) formed by C-mannosylated Trp5 and Trp8 with adjacent residues, calculated using the VMD Hydrogen Bonds plugin. Oxygen atoms are depicted in red and nitrogen atoms in dark-blue. (D) Most abundant hydrogen bonds formed during each simulation either by C-mannoses together with corresponding tryptophan residues or by non-mannosylated tryptophans alone are displayed as occupancies (%). A more detailed representation is depicted in *Figure 4—figure supplement 4*.

*Figure 4 continued on next page*

*Figure 4 continued*

The online version of this article includes the following video, source data, and figure supplement(s) for figure 4:

**Source data 1.** RMSF values from MD simulations at elevated temperatures.
**Figure supplement 1.** Side chain Root-Mean-Square Fluctuations (RMSF) of all residues from molecular dynamics simulations at elevated temperature.
**Figure supplement 2.** Cation-π interactions between Trp and Arg residues in the Trp-Arg ladder of the C-mannosylated (black) and non-mannosylated (blue) UNC-5 TSR2 during molecular dynamics simulation at elevated temperature.
**Figure supplement 3.** Molecular dynamics simulations of C-mannosylated UNC-5 TSR2 at elevated temperature with C-mannoses in the $^1C_4$ conformation.
**Figure supplement 4.** Total hydrogen bonds formed by C- and non-mannosylated Trp5 (**A**) and Trp8 (**B**) during molecular dynamics simulations at elevated temperature.
**Figure 4—video 1.** Exemplary, second simulation replicate of non-mannosylated TSR2 at 75˚C.
https://elifesciences.org/articles/52978#fig4video1
**Figure 4—video 2.** Exemplary, first simulation replicate of C-mannosylated TSR2 at 75˚C, with C-mannoses in the $^4C_1$ conformation.
https://elifesciences.org/articles/52978#fig4video2
**Figure 4—video 3.** Simulation of C-mannosylated TSR2 at 75˚C with C-mannoses in the $^1C_4$ conformation.
https://elifesciences.org/articles/52978#fig4video3

simulations (*Figure 4C,D* and *Figure 4—figure supplement 4*). Thus, C-mannoses appeared to keep the tryptophans in specific orientations not only by affecting their conformational space but also by forming polar interactions with surrounding residues.

We performed the simulations with mannoses in a $^4C_1$-chair conformation according to previously published crystal structures of C-mannosylated TSRs (*Aleshin et al., 2012*; *Hamming et al., 2012*). However, because NMR studies (*de Beer et al., 1995*; *Nishikawa et al., 2005*) and recently resolved TSR crystal structures (*Pedersen et al., 2019*; *Pronker et al., 2016*; *van den Bos et al., 2019*) suggest that C-mannoses can occur in $^4C_1$ and $^1C_4$ forms, we also performed a simulation with C-mannoses in $^1C_4$-chair arrangement (*Figure 4—video 3*). For both conformations, we observed similar maintenance of the Trp-Arg ladder and ability to form hydrogen bonds (*Figure 4—figure supplements 3* and *4*).

## C-mannosylation decreases the unfolding rate of UNC-5 TSR2 under reducing conditions

To evaluate the effects of C-mannosylation on the structure of a TSR aside from the impact exerted by the disulfide bonds, we analyzed non- and C-mannosylated UNC-5 TSR2 under reducing conditions by CD spectroscopy. Upon addition of 2 mM DTT, both proteins started to unfold immediately, showing that the disulfides are essential for the TSR structure. Denaturation of the C-mannosylated TSR, however, occurred approximately 1.8 times slower than that of the non-mannosylated repeat (*Figure 5A*). C-mannosylation might decrease the accessibility of the disulfide bridges by DTT, but could also protect the Trp-Arg ladder organization of the TSR.

To visualize putative structural effects of C-mannosylation, we performed molecular dynamics simulation studies of non- and C-mannosylated UNC-5 TSR2 lacking disulfide bonds at 24˚C. As in the thermal denaturation simulation, non-mannosylated TSR revealed an increased flexibility of Trp5 in the Trp-Arg ladder organization in comparison to the C-mannosylated TSR (*Figure 5B,C* and *Figure 5—figure supplement 1*). Additionally, Cys26 (marked by *) involved in the upper disulfide bridge showed lower fluctuation values and thus decreased flexibility in the C-mannosylated TSR (*Figure 5C*, *Figure 5—figure supplement 1*) which might be relevant for the disulfide bond formation during protein folding.

## C-mannosylation supports folding of UNC-5 TSR2

To further investigate the role of C-mannoses in TSR folding, oxidative folding experiments were performed with non- and di-mannosylated UNC-5 TSR2. After reduction and unfolding of the TSRs with DTT and guanidine hydrochloride, the refolding of the proteins was monitored by CD spectroscopy in the presence of 1 mM reduced and 0.5 mM oxidized glutathione (GSH and GSSG) for three hours. Both TSRs were able to form native-like conformations as demonstrated by the appearance of the characteristic TSR peak in CD spectra (*Figure 6A*). Still, folding initiation of the C-mannosylated

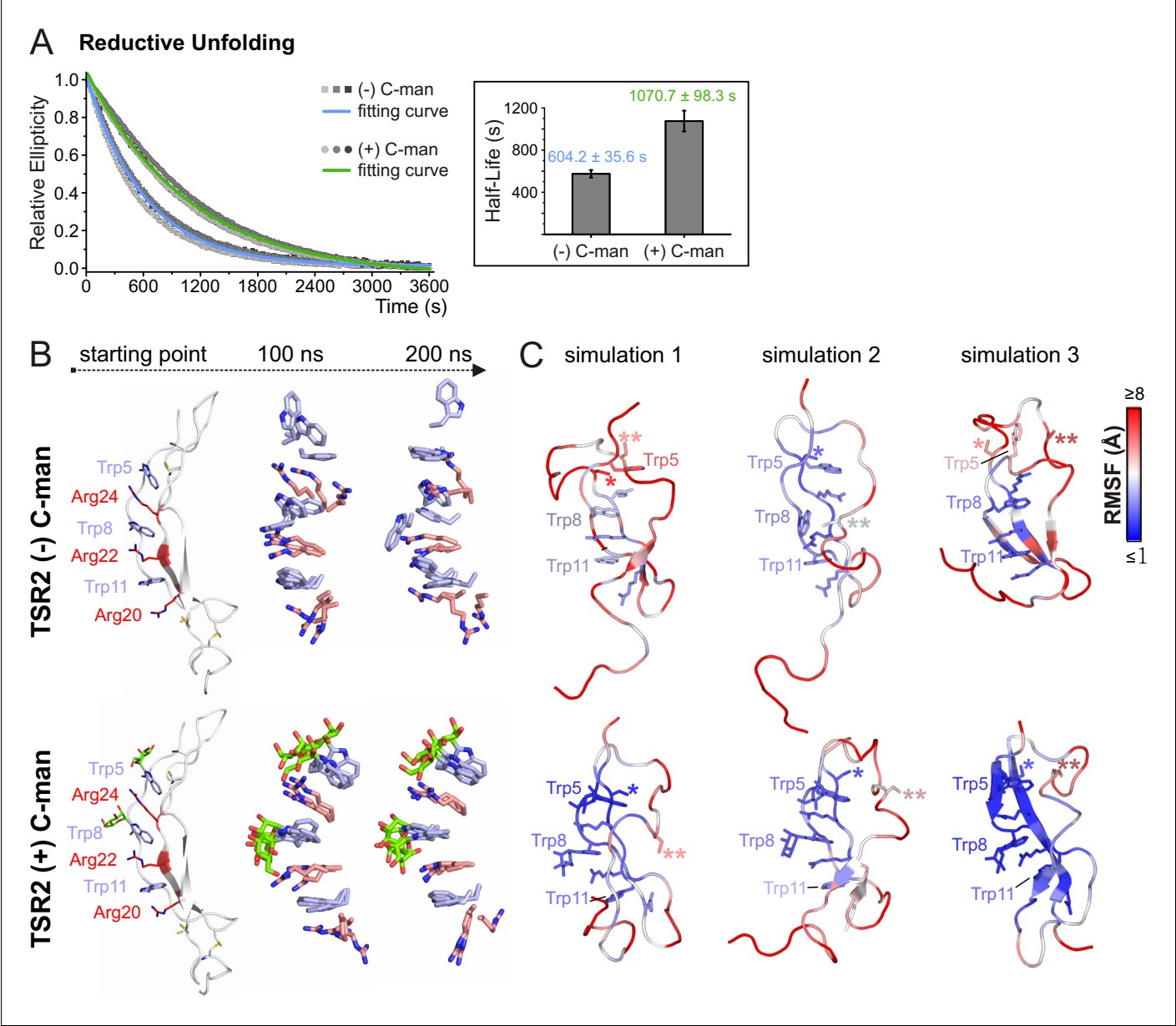

**Figure 5.** C-mannosylation decreases the unfolding rate of UNC-5 TSR2 during reductive denaturation. (**A**) Reductive denaturation of non-mannosylated (blue) and C-mannosylated (green) UNC-5 TSR2 monitored using CD spectroscopy at a wavelength of 229 and 228 nm, respectively. Measurements of three non-mannosylated (gray-shaded squares) and C-mannosylated (gray-shaded circles) UNC-5 TSR2 samples were averaged and fitted (*Figure 5—source data 1*). Average half-life values are depicted in the right plot (error bars show standard deviation). (**B**) Modeled structures of UNC-5 TSR2 lacking disulfide bridges without C-mannosylation (top) and with two C-mannoses on Trp5 and Trp8 (bottom) at the starting point of each simulation followed by aligned Trp-Arg ladder structures from three simulation replicates after 100 and 200 ns simulation time. Tryptophans are depicted in light-blue, mannoses in green and arginines in salmon. (**C**) Final structures of non- and C-mannosylated UNC-5 TSR2 after each 200 ns simulation, colored using a blue-white-red gradient according to the calculated root-mean-square fluctuation (RMSF) values (blue ≤1 Å; red ≥5 Å) (*Figure 5—figure supplement 1*, *Figure 5—source data 2*). Cys26 and Cys38 residues, that are involved in the upper disulfide bridge of the TSR, are labeled with one or two asterisks, respectively.

The online version of this article includes the following source data and figure supplement(s) for figure 5:

**Source data 1.** Reductive denaturation of C- and non-mannosylated UNC-5 TSR2.
**Source data 2.** RMSF values from MD simulations of TSR2 lacking disulfide bridges.
**Figure supplement 1.** Side chain Root-Mean-Square Fluctuations (RMSF) of all residues from molecular dynamics simulations of the TSRs lacking disulfide bonds.

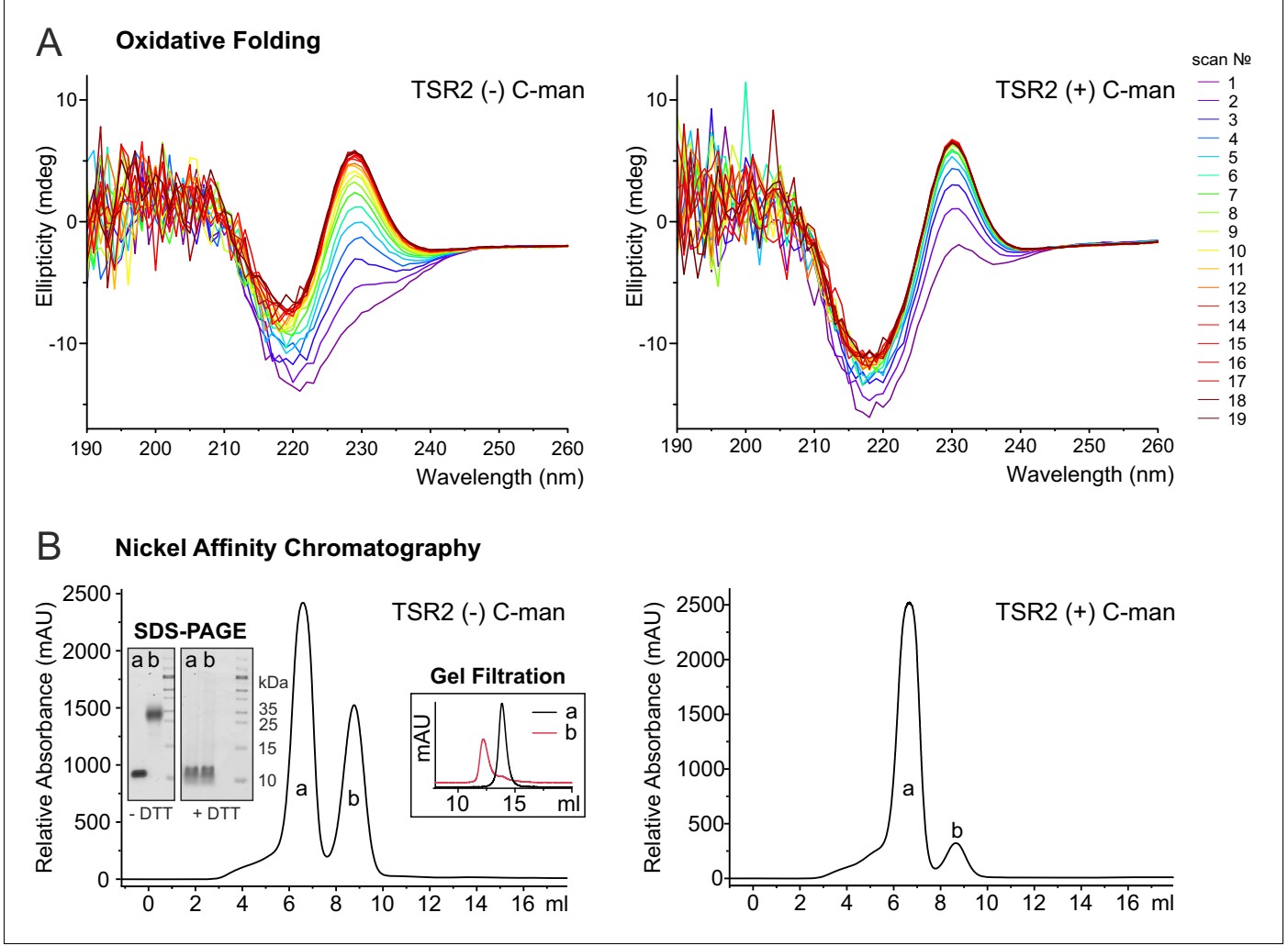

**Figure 6.** C-mannosylation impacts the folding of UNC-5 TSR2. (**A**) Oxidative refolding of previously reduced and unfolded non- and di-mannosylated UNC-5 TSR2 monitored by CD spectroscopy in presence of 0.5 mM GSSG and 1 mM GSH at 24°C. The first spectrum was recorded 9 min after removal of denaturing agents and addition of GSSG/GSH. Subsequent spectra were recorded every 10 min. (**B**) Nickel affinity purification chromatograms of secreted non- and C-mannosylated His-tagged TSR2 from *Drosophila* S2 cells. Whereas C-mannosylated TSR2 appeared primary as monomer (**a**), non-mannosylated TSR revealed a high amount of dimeric structures (**b**), which bound stronger to the nickel affinity column and were confirmed by gel filtration chromatography and non-reducing SDS-PAGE. Highly increased formation of dimers was consistently observed in over ten independent purifications.

The online version of this article includes the following figure supplement(s) for figure 6:

**Figure supplement 1.** Time plot of the oxidative refolding presented in *Figure 6A*.

**Figure supplement 2.** MALDI-TOF MS analysis of monomeric and dimeric UNC-5 TSR2 structures.

**Figure supplement 3.** Separation of di-, mono- and non-mannosylated UNC-5 TSR2 by C18 chromatography.

---

TSR occured significantly faster compared to the non-modified protein (*Figure 6A*, *Figure 6—figure supplement 1*), confirming that C-mannosylation plays a role in the folding of UNC-5 TSR2.

Indeed, purification of non- and C-mannosylated TSRs from *Drosophila* S2 cells showed that the folding of non-mannosylated TSRs was affected. Next to monomeric TSRs, a fraction with higher nickel affinity was secreted (*Figure 6B*). This fraction was interpreted as a covalently linked dimer since it showed a higher mobility in gel filtration (*Figure 6B*) and the double mass of the monomer in the MALDI-TOF spectrum (*Figure 6—figure supplement 2*). On SDS-PAGE, the mobility became identical to the monomer upon reduction by DTT (*Figure 6B*) indicating connection by aberrant disulfides. In cells co-expressing DPY-19 C-mannosyltransferase, dimeric TSR formation was

significantly reduced. These results argue for an accelerated folding initiation of the C-mannosylated TSR in the ER and a critical role of C-mannosylated tryptophans for the native disulfide connections.

## Discussion

C-mannosylation is a unique protein glycosylation type with a still unclear function. In the present study, we compared non- and C-mannosylated UNC-5 TSRs and were able to show that C-mannoses added on the first two tryptophans of the WXXWXXW motif affect the dynamics of a TSR and play an important role in its folding, stability and secretion.

Assuming that C-mannosylation is co-translational (*Krieg et al., 1998*) implies that it precedes and thereby assists correct Cys pairing to form the native disulfide bridges. Indeed, we observed increased formation of covalent dimers (*Figure 6B*) for TSR2 of *C. elegans* UNC-5 in the absence of C-mannosylation, which were caused by incorrect disulfide bond connections. Our molecular dynamics simulations of UNC-5 TSR2 lacking disulfide bridges showed that the Trp-Arg ladder was maintained when C-mannosylation was present (*Figure 5B*). Accordingly, we observed slower unfolding of the C-mannosylated TSR during protein denaturation under reductive conditions (*Figure 5A*). The simulation also showed that Cys26 fluctuated less if the tryptophans were C-mannosylated. In the process of protein folding, cysteines have to come in close vicinity to form bridges. The Trp-Arg ladder most likely plays an essential role in the correct positioning of cysteines. It is conceivable that the mannoses assist in the formation of the ladder and stabilize it before disulfide bridges are made. Our oxidative folding experiments indeed showed that the refolding initiation (appearance of the Trp-Arg ladder) was accelerated by the presence of C-mannoses (*Figure 6A*). This is also consistent with the view that protein domain folding precedes disulfide bridge formation (*Robinson et al., 2017*).

Once the TSR is folded, it is further stabilized by the three disulfide bridges, and although the shift of the maximum in the CD spectrum suggests that the tryptophan and arginine residues are in slightly different environments, the overall CD profiles are comparable in both TSR forms (*Figure 3A*). Nevertheless, C-mannosylated TSR was more resistant to thermal denaturation than non-mannosylated TSR (*Figure 3B*) revealing the importance of C-mannosylation also after folding of the TSR. Molecular dynamics simulations at elevated temperature showed that C-mannoses were able to built hydrogen bonds to the surrounding residues (*Figure 4C,D*) and kept especially the tryptophan closest to the N-terminus in place. Without C-mannose, this tryptophan was consistently losing its proper orientation in the ladder in repeated simulations, resulting in higher flexibility of the upper TSR region (*Figure 4A,B*). Existence of hydrogen bonds between C-mannoses and surrounding amino acids was also suggested in recent structural studies on properdin (*Pedersen et al., 2019*; *Pronker et al., 2016*; *van den Bos et al., 2019*).

The area around the third tryptophan was less affected. Since the *C. elegans* C-mannosyltransferase DPY-19 is not targeting this tryptophan, it remains unmodified in *C. elegans* UNC-5. In mammals, the third tryptophan of the WXXWXXWXXC sequence can be mannosylated by the DPY19L3 homolog (*Hofsteenge et al., 1999*; *Shcherbakova et al., 2017*). The requirement for C-mannosylation at this position, as well as on the first two tryptophans, might thus vary in different organisms and different proteins. Secretion of human R-spondin1 was highly reduced in DPY19L3-restricted cells (*Niwa et al., 2016*), whereas mouse UNC5A was not affected (*Shcherbakova et al., 2017*). Moreover, many TSRs have less than three tryptophans, and some might not be C-mannosylated at all (*Verbij et al., 2016*). In *C. elegans* MIG-21, the first TSR lacks two of the six cysteines and might, therefore, require C-mannosylation since it cannot be secreted without C-mannoses even at low temperatures (*Buettner et al., 2013*). On the other hand, multiple TSRs have been produced in native conformations in bacteria or insect cells without glycosylation (*Klenotic et al., 2011*; *Tan et al., 2002*; *Tossavainen et al., 2006*), showing that C-mannosylation is probably not an absolute requirement for some TSRs, but a critical factor for others.

*C. elegans dpy-19* mutants deficient in C-mannosylation develop a temperature-sensitive dumpy phenotype (*Honigberg and Kenyon, 2000*) that we could reproduce (*Figure 2—figure supplement 1*). We assume that in these mutants, secretion and stability of various TSR-containing and possibly other proteins are affected. Since impairment of UNC-5 alone is not sufficient for the development of the dumpy phenotype (*Hedgecock et al., 1990*), further C-mannosylation-dependent proteins must be involved.

In conclusion, our experiments support a model in which C-mannosylation of tryptophans in the Trp-Arg ladder of TSRs can provide assistance in the folding process, but also greatly enhance the stability of the folded protein.

# Materials and methods

## Key resources table

| Reagent type (species) or resource | Designation | Source or reference | Identifiers | Additional information |
|---|---|---|---|---|
| Cell line (*D. melanogaster*) | S2 | Thermo Fisher | S2 (Schneider 2), R69007 | |
| Strain, strain background (*C. elegans*) | *C. elegans dpy-19* mutants | *Caenorhabditis* Genetics Center | *dpy-19* (e1259) III, strain CB1259 | |
| Recombinant DNA reagent | pIB-DPY-19 | (*Buettner et al., 2013*) | | |
| Recombinant DNA reagent | pMT-UNC-5-TSR1+2 | this paper | | *Supplementary file 1* |
| Recombinant DNA reagent | pMT-UNC-5-TSR2 | this paper | | *Supplementary file 1* |
| Recombinant DNA reagent | pMT-Notch-EGF16–20 | this paper | | *Supplementary file 1* |
| Antibody | anti-V5 (mouse monoclonal) | Acris | SM1691 | WB (1:5000) |
| Software, algorithm | MODELLER | (*Sali and Blundell, 1993*) | RRID: SCR_008395 | |
| Software, algorithm | Schrödinger MacroModel | Schrodinger Suite, available online (*Watts et al., 2014*) | RRID: SCR_016747 | |
| Software, algorithm | NAMD 2.11 | (*Phillips et al., 2005*) | RRID: SCR_014894 | |
| Software, algorithm | CHARMM27 force field | (*MacKerell et al., 1998*) | | |
| Software, algorithm | Particle mesh Ewald method | (*Darden et al., 1993*) | | |
| Software, algorithm | data2bfactor, color_b scripts | The PyMOL Molecular Graphics System, Version 1.2r3pre, Schrödinger, LLC | RRID: SCR_000305 | |
| Software, algorithm | hierarchical clustering | (*Kelley et al., 1996*) | RRID: SCR_004097 | |
| Software, algorithm | CaPTURE | (*Gallivan and Dougherty, 1999*) | | |
| Other | PiStar-180 | Applied Photophysics | | |

## Plasmids and protein constructs

The pIB vector (Invitrogen) was used for constitutive expression of the *C. elegans* DPY-19 protein (pIB-DPY-19) (*Buettner et al., 2013*). The pMT/BiP/V5-His A vector (Invitrogen) was used for expression of V5- and His-tagged *C. elegans* UNC-5 TSRs 1+2 and *Drosophila* Notch EGF repeats as well as His-tagged *C. elegans* UNC-5 TSR2 (corresponding protein sequences are depicted in *Supplementary file 1*).

## Protein expression in S2 cells for western blot analysis

Transfection of *Drosophila* S2 cells in Insect-XPRESS Protein-free Medium (Lonza) was performed using FuGENE HD (Promega) and Opti-MEM (Invitrogen) – 2 ml cells were transfected using 2 µg

pMT-UNC-5-TSR1+2 plasmid and 1 µg pMT-Notch-EGF16–20. After 6 hr at 24°C, pMT-vector expression was induced with 0.2 mM CuSO$_4$. Induced cells were sub-divided and incubated at 20, 24°C and 28°C for 3 days. 1/10[th] of the cell media was mixed with 5 × Laemmli (300 mM Tris-HCl [pH 6.8], 10% SDS, 50% glycerol, 25% β-mercaptoethanol, 0.05% bromophenol blue). 1/10[th] of cells was resuspended in 20 µl 1 × Laemmli. All samples were heated for 10 min at 95°C and separated by SDS-PAGE (5%/15%). Proteins were detected on a nitrocellulose membrane with mouse anti-V5 antibody (1:5000; Acris) and IRDye 800 conjugated goat anti-mouse secondary antibody (1:20000; LI-COR). Blots were scanned on a LI-COR Odyssey Infrared Scanner. Prestained marker bands (Page-Ruler, Fermentas) were detected at 700 nm. The experiment was repeated five times with independently transfected cells.

## Protein expression in S2 cells for CD spectroscopy

20 ml S2 cells were transfected either with 20 µg pMT-UNC-5-TSR2 plasmid or with 10 µg pMT-UNC-5-TSR2 and 10 µg pIB-DPY-19 as described above. Upon induction, cells expressing UNC-5 TSR2 were incubated at 24°C, and cells expressing UNC-5 TSR2 and DPY-19 at 28°C (to avoid secretion of non-mannosylated UNC-5 TSR2). After 4 days of incubation, cell media were centrifuged for 3 min at 300 × g and 4.500 × g, filtered through 0.2 µm cellulose acetate membrane (Waters) and dialyzed against 500 ml 20 mM Tris-HCl pH 8, 500 mM NaCl using 2000 MWCO dialysis tubes (Carl Roth). For nickel affinity chromatography, samples were mixed with 20 mM imidazole and loaded on 1 ml HisTrap HP columns (GE Healthcare). Proteins were eluted with a linear gradient of 20–350 mM imidazole over 7 ml and detected at 280 nm. Fractions containing monomeric UNC-5 TSR2 were applied to a 30 ml gel filtration column (Bio-Gel P-10 fine, Bio-Rad) in 10 mM KP$_i$ pH 7.6, 5 mM NaCl. Protein-containing fractions were concentrated using Vivaspin 6 MWCO 3000 (GE Healthcare).

## Protein expression in S2 cells for oxidative folding

200 ml of stably transfected S2 cells expressing either pMT-UNC-5-TSR2 alone or in combination with pIB-DPY-19 at a density of $3 \times 10^6$ cells/ml were induced using 4 µM CdCl$_2$ and incubated for 4 days at 24 or 28°C, respectively. The cell media were centrifuged, filtered, dialyzed and purified by nickel affinity chromatography as described above. The monomeric proteins were additionally purified by C18 chromatography (Xbridge TM Prep C18 5 µm, 10 × 50 mm column, Waters) to separate di-, mono- and non-mannosylated TSR2 forms (*Figure 6—figure supplement 3*) prior to gel filtration in 100 mM Tris pH 8 and concentration.

## CD spectroscopy, thermal and reductive denaturation

Measurements were performed with a 3 mm high precision quartz cuvette (Hellma Analytics) using protein solutions of 0.1 mg/ml (in 10 mM KP$_i$ pH 7.6, 5 mM NaCl) in a PiStar-180 (Applied Photophysics) system with the Equilibrium Sampling Handling Unit (ESHU). To obtain native CD spectra, scans with 5 nm bandwidth of 260–178 nm with 1 nm steps and 16 s/step sampling time were performed at 24°C. Three spectra were averaged and smoothed using the Savitsky-Golay algorithm (3-point smooth). For thermal denaturation, 24–95°C temperature ramping with 1°C steps and 1.5 °C/min ramping rate was used. For reductive denaturation, 2 mM DTT was added to the samples, and after 2 min mixing time, the kinetics were detected with 16 s/step sampling time for 3600 s at 24°C. In both denaturation experiments, the ellipticity changes were monitored at 228 nm for the C-mannosylated and at 229 nm for non-mannosylated TSR. To determine the melting temperatures, the average values from measurements of three protein samples, independently produced by transient transfection of S2 cells, were calculated and fitted by a Boltzmann function (OriginLab). For calculations of the half-time values, a first-order exponential decay function was used for the fitting (ExpDec1, OriginLab).

## Molecular dynamics simulations

A structural model of UNC-5 TSR2 was based on the available crystal structure of the human UNC5A TSR (PDB 4V2A, 62% amino acid positives and no gaps) using MODELLER (*Sali and Blundell, 1993*). Positions of C-mannoses on Trp5 and Trp8 were derived from the positions of C-mannoses in complement component C6 (3T5O), IL-21 receptor (3TGX) and ADAMTS13 (3VN4). Force field parameters for C-mannoses were created manually. Prior to MD simulations, the structural models were

energy minimized using Schrödinger MacroModel and the OPLS3 force field (Schrodinger Suite, available online; *Watts et al., 2014*), and fully solvated with the TIP3P water model (*Jorgensen et al., 1983*). The net charge of the systems was neutralized by adding counter ions. All simulations were performed using NAMD 2.11 (*Phillips et al., 2005*) and the CHARMM27 force field (*MacKerell et al., 1998*). A time-step of 1 fs was used. Long-range electrostatics were treated with the particle mesh Ewald method (*Darden et al., 1993*) and a 12 Å cutoff was used for nonbonded short-range interactions. First, an additional energy minimization and a 5 ns equilibration were performed at a constant temperature of 310 K and pressure (1 atm), followed by 200 ns MD simulations. For the simulation at elevated temperature, mimicking thermal denaturation conditions, MD simulations were carried out at constant temperatures of 348 K for 200 ns. For the simulation of the reductive denaturation, all disulfide bonds between cysteine residues were removed. The simulations were performed for 200 ns at 297 K. For both conditions, three independent simulations were performed and compared.

Hydrogen bonds between tryptophan residues and the protein were determined at distances below 3.5 Å and the angle cutoff of 30°. RMSF values of side chain atoms were calculated using an RMSF script, averaged for each residue and used in PyMOL for the coloring of the protein structures (data2bfactor script, color_b script, The PyMOL Molecular Graphics System, Version 1.2r3pre, Schrödinger, LLC). Clusters of most frequent conformations of the Trp-Arg ladder were determined by hierarchical clustering (*Kelley et al., 1996*) of the simulation trajectories at elevated temperatures. The calculated representative structures from the largest cluster of each simulation were used for RMSF-coloring. For the simulation of reducing conditions, the last structure of each simulation (after 200 ns) was used to demonstrate the extent of denaturation in non- and C-mannosylated TSRs.

For the calculation of putative cation-π interactions in the protein structures, PDB files of single frames from the MD trajectories were analyzed with the CaPTURE program (*Gallivan and Dougherty, 1999*).

## Oxidative folding

Non- and C-mannosylated UNC-5 TSR2 (0.14 mg/ml) were denatured in 200 mM DTT and 6 M guanidine hydrochloride overnight at 24°C. The samples were desalted at 32°C using a 5 ml HiTrap desalting column (GE Healthcare) and the folding buffer (50 mM Tris HCl pH 8.4, 100 mM NaCl). The refolding reaction was performed in the presence of 0.5 mM GSSG and 1 mM GSH (Sigma) at 24°C and was monitored for 3 hr by CD spectroscopy. Scans of 260–190 nm were performed with 1 nm steps and 8 s/step sampling time, whereby 10 min per scan were required. The experiment was performed four times in total, at two temperatures (24°C and 32°C), whereby the same effect was observed.

## Additional information

### Funding

| Funder | Grant reference number | Author |
|---|---|---|
| Deutsche Forschungsgemeinschaft | FOR2509 BA 4091/6-1 | Hans Bakker |
| Deutsche Forschungsgemeinschaft | BA 4091/5-1 | Hans Bakker |
| Deutsche Forschungsgemeinschaft | BU 2920/2-1 | Falk FR Buettner |

The funders had no role in study design, data collection and interpretation, or the decision to submit the work for publication.

### Author contributions

Aleksandra Shcherbakova, Conceptualization, Formal analysis, Validation, Investigation, Visualization, Methodology, Writing - original draft, Project administration, Writing - review and editing; Matthias Preller, Conceptualization, Resources, Data curation, Software, Formal analysis, Supervision, Investigation, Visualization, Methodology, Writing - review and editing; Manuel H Taft, Resources,

Software, Methodology, Writing - review and editing; Jordi Pujols, Visualization, Methodology, Writing - review and editing; Salvador Ventura, Supervision, Methodology, Writing - review and editing; Birgit Tiemann, Formal analysis, Investigation, Methodology; Falk FR Buettner, Resources, Software, Formal analysis, Methodology, Writing - review and editing, Funding; Hans Bakker, Conceptualization, Supervision, Funding acquisition, Investigation, Methodology, Writing - original draft, Project administration, Writing - review and editing

### Author ORCIDs
Aleksandra Shcherbakova  https://orcid.org/0000-0003-4175-547X
Matthias Preller  http://orcid.org/0000-0002-7784-4012
Manuel H Taft  http://orcid.org/0000-0001-5853-8629
Salvador Ventura  http://orcid.org/0000-0002-9652-6351
Falk FR Buettner  https://orcid.org/0000-0002-8468-1223
Hans Bakker  https://orcid.org/0000-0002-1364-9154

### Decision letter and Author response
Decision letter https://doi.org/10.7554/eLife.52978.sa1
Author response https://doi.org/10.7554/eLife.52978.sa2

## Additional files

### Supplementary files
• Supplementary file 1. Sequences of used *C. elegans* UNC-5 and *Drosophila* Notch constructs.
• Transparent reporting form

### Data availability
All data generated or analysed during this study are included in the manuscript and supporting files. Source data files have been provided for Figures 3, 4 and 5.

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
