## [Decision Letter]

**Acceptance summary:**

C-mannosylation of tryptophan is a relatively newly discovered and still understudied post-translational modification. Using a well-controlled system and side-by-side comparison of glycosylated vs. non-glycosylated thrombospondin type I repeats of netrin receptor UNC-5, this study provides a comprehensive assessment of the effects of C-mannosylation on protein folding and on the ability of the protein to withstand thermal and reductive denaturation. Despite the inherent strength of cation-π interactions, it appears that C-mannosylation of tryptophans in a tryptophan-arginine ladder provides further stabilization of this motif, and thereby of the domain.

**Decision letter after peer review:**

Thank you for submitting your article "C-Mannosylation supports folding and enhances stability of thrombospondin repeats" for consideration by *eLife*. Your article has been reviewed by three peer reviewers, including Deborah Fass as the Reviewing Editor and Reviewer #1, and the evaluation has been overseen by David Ron as the Senior Editor. The following individual involved in review of your submission has agreed to reveal their identity: Tadashi Suzuki (Reviewer #2).

The reviewers have discussed the reviews with one another and the Reviewing Editor has drafted this decision to help you prepare a revised submission.

C-mannosylation is an unusual form of glycosylation observed on tryptophans in certain WXXW motifs such as those found in thrombospondin type 1 repeats. This manuscript describes a biochemical and in silico comparison of two versions of a TSR-containing protein, one with and one without C-mannose modifications. The authors show that C-mannosylation affects thermal denaturation, denaturation induced by reductant, and the rate of oxidative folding. It was previously shown that C-mannosylation can affect expression levels of proteins, but a direct comparison of purified proteins in vitro is novel.

Summary:

This study convincingly demonstrates that C-mannosylation can affect thermal and denaturant resistance of proteins, and that it can alter the efficiency of oxidative protein folding.

Essential revisions:

The authors use the term "stability" in a non-rigorous manner. Stability of proteins refers to the difference in free energy between the folded and unfolded states and thus relates to thermodynamics. In the current manuscript, the experiments monitor the resistance to thermal denaturation and the kinetics of unfolding upon reduction, neither of which provides data on thermodynamic parameters.

Is the thermal denaturation reversible?

In Figure 3, if "three independent samples" refers to technical replicates, this should be stated. Also, a standard deviation seems more appropriate than a standard error here.

Figure 4D was not mentioned in the text.

Regarding Figure 5A, the glycan may simply be hindering the accessibility of the added reductant to the disulfide bonds. The authors should also keep in mind that their experiment does not distinguish between the rate of reduction and the rate of denaturation upon reduction.

The data in Figure 6A should be plotted also as the 228/229 nm peak height as a function of time after initiating folding, and it should be made clear if the total recovery of folded protein is the same for both versions.

What are the shoulders in the 1 mannose and 2 mannoses peaks in Figure 6—figure supplement 3?

---

## [Author Response]

Essential revisions:The authors use the term "stability" in a non-rigorous manner. Stability of proteins refers to the difference in free energy between the folded and unfolded states and thus relates to thermodynamics. In the current manuscript, the experiments monitor the resistance to thermal denaturation and the kinetics of unfolding upon reduction, neither of which provides data on thermodynamic parameters.

We indeed used the term “stability” in a more broad and general manner. Terms like “thermal stability of proteins” or “protein stabilization by disulfide bridges or cation-π interactions” are, however, well-accepted in use. Thus, we thought that we could designate the effects of C-mannosylation as generally stabilizing if we had good arguments for this.

Still, we acknowledge your point that it is formally inaccurate and either replaced the term “stability” by the suggested term “resistance”, tried to specify it more precisely or paraphrased the concerned text passages. We only left the term when we speak in general of e.g. “impact on protein folding and stability”.

Is the thermal denaturation reversible?

Yes, the thermal denaturation is reversible to a high extent in non- and Cmannosylated TSRs. We included the corresponding data as Figure 3—figure supplement 2 with a reference in the Figure 3B legend.

In Figure 3, if "three independent samples" refers to technical replicates, this should be stated. Also, a standard deviation seems more appropriate than a standard error here.

The three samples have been independently produced by transient transfection of *Drosophila* S2 cells. We do not consider this technical replicates, but it cannot be called biological replicates either. We have removed the term “independent” in the figure legends and described the independence in Materials and methods – CD spectroscopy, thermal and reductive denaturation: “To determine the melting temperatures, the average values from measurements of three protein samples, independently produced by transient transfection of S2 cells, were calculated and fitted by a Boltzmann function (OriginLab).”

Standard error was replaced by standard deviation in Figures 3B and 5A.

Figure 4D was not mentioned in the text.

Figure 4D assignment is now included in the main text: Results subsection “Molecular dynamics simulation of the thermal denaturation”; Discussion.

Regarding Figure 5A, the glycan may simply be hindering the accessibility of the added reductant to the disulfide bonds. The authors should also keep in mind that their experiment does not distinguish between the rate of reduction and the rate of denaturation upon reduction.

We included that point in the main text: “C-mannosylation might decrease the accessibility of the disulfide bridges by DTT, but could also protect the Trp-Arg ladder organization of the TSR.”

The data in Figure 6A should be plotted also as the 228/229 nm peak height as a function of time after initiating folding, and it should be made clear if the total recovery of folded protein is the same for both versions.

We included the time plot with four oxidative folding reactions of non- and C-mannosylated TSRs in Figure 6—figure supplement 1, where the recovery can be compared between the proteins after three hours.

In our eyes, the faster initiation of the folding in case of the C-mannosylated TSR is the most crucial aspect of that experiment since it may play an important role during the TSR folding in the ER. We tried to make that point more evident in the main text and adapted the corresponding results part accordingly.

What are the shoulders in the 1 mannose and 2 mannoses peaks in Figure 6—figure supplement 3?

The shoulder appears in all three glycoforms (0, 1, and 2 Man). It most likely arises from an unknown additional protein modification. In peptide MS, various modifications, like oxidation, are observed, but we are not able to address a specific modification to the shoulder seen in C18 chromatography.